# Effects of Sugarcane/Peanut Intercropping on Root Exudates and Rhizosphere Soil Nutrient

**DOI:** 10.3390/plants13223257

**Published:** 2024-11-20

**Authors:** Xiumei Tang, Lulu Liao, Haining Wu, Jun Xiong, Zhong Li, Zhipeng Huang, Liangqiong He, Jing Jiang, Ruichun Zhong, Zhuqiang Han, Ronghua Tang

**Affiliations:** 1Cash Crops Research Institute, Guangxi Academy of Agricultural Sciences, Nanning 530000, China; tangxiumei2005@163.com (X.T.); liaolulu1007@163.com (L.L.); wuhaining86@foxmail.com (H.W.); xiongjun9065@sina.com (J.X.); zhjxc15@163.com (Z.H.); heliangqiong@163.com (L.H.); jiangjing_gx@163.com (J.J.); rczhong6501@163.com (R.Z.); hanzhuqiang@163.com (Z.H.); 2Guangxi Crop Genetic Improvement and Biotechnology Lab, Nanning 530007, China; gxlizhong@126.com

**Keywords:** intercropping, root metabolites, root exudates, rhizosphere soil enzymes, rhizosphere soil nutrients

## Abstract

Intercropping can enable more efficient resource use and increase yield. Most current studies focus on the correlation between soil nutrients and crop yield under intercropping conditions. However, the mechanisms related to root exudates and soil nutrients remain unclear. Therefore, this study explored the correlation between rhizosphere soil nutrients and root exudates in sugarcane/peanut intercropping. Root extracts, root exudates, rhizosphere soil enzyme activities, and soil nutrients were analyzed and compared in monocultured and intercropped peanut and sugarcane at different growth stages. The root metabolites were annotated using the Kyoto Encyclopedia of Genes and Genomes pathways to further identify the connection between soil nutrients and root exudates. The effects of intercropping differed in peanut and sugarcane at different growth stages, and the difference between podding and pod-filling stages was significant. Intercropping generally had a great effect on peanut; it not only significantly increased the organic acid, soluble sugars, and phenolic acids in root exudates and extracts from peanuts, but also significantly increased rhizosphere soil enzyme activities and soil nutrient levels. Intercropping peanuts promoted fumaric acid secretion from roots and significantly affected the metabolic pathways of alanine, aspartate, and glutamate. Sugarcane/peanut intercropping can increase root exudates and effectively improve soil nutrients. The changes in soil nutrients are closely related to the effects of fumaric acid on alanine, aspartate, and glutamate metabolism.

## 1. Introduction

Sugarcane and peanut are the main cash crops and oil crops in Guangxi, China, and they play an important role in the economic construction of Guangxi. However, long-term continuous cropping of peanut or sugarcane may lead to the deterioration of soil quality, the intensification of diseases and pests, the weakening of ecological balance, and the reduction in crop yield and quality [1,2]. The bare time between sugarcane rows is consistent with the peanut growth period, and the temperature and humidity requirements for sugarcane and peanut planting are similar. According to the mutualism principle and biological characteristics of crops, a reasonable sugarcane/peanut intercropping pattern has been established that not only alleviates the obstacles of continuous cropping, but also fully utilizes resources, increases production and efficiency, and combines land with cultivation [3,4,5].

Intercropping has been widely practiced and applied worldwide, especially in developing countries. Intercropping can improve the yield and quality of crops and promote sustainable agricultural development [6]. Intercropping promotes efficient interspecific nutrient utilization, a phenomenon that is more pronounced in legume and grass intercropping [7,8]. The direct effect of legume and grass intercropping is to increase resource availability, and the indirect effect is to promote ecosystem differentiation and complementarity, reduce ecosystem overlap and competition, and the combination of the two is effective in increasing yield [9,10]. For example, maize and soybean intercropping increased photosynthetically active radiation and light energy utilization at the top of the soybean canopy [11], and sugarcane and peanut intercropping increased microbial diversity in rhizosphere soil [12].

Intercropping can promote root secretion of organic compounds [13],and the organic compounds released are key factors in the mineralization of acquired nutrients and mediating plant–microbial interactions. Root exudates and soil–root interactions affect soil nutrient dynamics and microbial communities and indirectly affect plant growth and development [14]. Root exudates are the mediators of material exchange, energy flow, and information transfer, and play a key role in alleviating stressors that are intimately associated with plant growth. The quantity and quality of root exudates are determined by external factors such as plant species, plant age, and biological and abiotic stress [15]. Intercropping effectively uses natural resources, increases yield, and minimizes pesticide and fertilizer use [4,16]. For example, sugarcane/peanut intercropping can improve the rhizosphere soil microecological environment by increasing the soil available nitrogen (N) content, microbial quantity, microbial diversity, and soil enzyme activity [17]. Maize/peanut intercropping increased alkali-hydrolyzed nitrogen (AN), available phosphorus (AP), and soil organic matter (SOM) contents, and the electrical conductivity (EC) in the rhizosphere soil varied to different degrees, with maize intercropping showing a more significant increase in soil nutrients [18]. Zhang et al. [19] reported that compared with monocroping, the maize/broad bean intercropping improves the phosphorus utilization rate of maize through rhizosphere citric acid and acid phosphatase secretion. The root exudates of maize not only contribute to the formation of root nodules of broad bean, but also stimulate the synthesis of flavonoids in the root, thus improving the nodulation ability of broad bean [20]. Song et al. [8] found that intercropping wheat with broad bean and maize increased the effectiveness of crop microbial biomass carbon, nitrogen, and phosphorus compared to monocropping. In summary, the current research on intercropping focuses more on soil nutrient changes and crop yield improvement, and the correlation mechanism between root secretion and soil nutrients is still unclear. Therefore, the study of root exudates and rhizosphere soil can reflect the interaction between two crop-growing periods.

In this study, the rhizosphere soil nutrients, root exudates, extracts, and metabolites were determined and analyzed with sugarcane/peanut intercropping at four growth stages to: (1) determine changes in soil nutrients, enzyme activities, root extracts, exudates, and metabolites under intercropping conditions; (2) identify the differential metabolites in peanut intercropping (IP) and sugarcane intercropping (IS); and (3) conduct correlation analysis of soil nutrients, enzyme activities, root extracts, exudates, and metabolites. This study offers a theoretical foundation for further exploration into achieving high quality and high yields of sugarcane and peanut through intercropping in southwest Guangxi.

## 2. Results

### 2.1. Effects of Sugarcane/Peanut Intercropping on the Organic Acid Content

Among the four growth periods in the root extracts (Figure 1a), the organic acid content at the seedling stage was significantly higher than that at the flowering-pegging, podding, and pod-filling stages. Compared to MP, IP significantly decreased the organic acid content from four stages, but the differences were not significant. Compared to MS, IS significantly decreased the organic acid content at the seedling stage. These results indicate that intercropping significantly decreased the organic acid content in root extracts of IS at the seedling stage.

In the comparison of the four growth stages in the root exudates (Figure 1b), the organic acid content of all treatments showed a trend of first decreasing and then increasing. The organic acid content at the podding, flowering-pegging, and pod-filling stages was significantly lower than that at the seedling stage, and the organic acid content was the lowest at the podding stage. Compared to the seedling stage, the organic acid content at the podding stage in all treatments was reduced by 78–80%. Compared to MP, the organic acid content at the seedling stage significantly increased in IP by 71%. Compared to MS, the organic acid content at the seedling stage and pod-filling stage in IS decreased significantly, by 15% and 29%, respectively. These results indicate that intercropping significantly increased the organic acids in the root exudates in IP at the seedling stage; in contrast, the opposite trend was observed in IS at the seedling and pod-filling stages.

### 2.2. Effects of Sugarcane/Peanut Intercropping on the Soluble Sugar Content

Comparing the four growth stages in the root extracts (Figure 2a), the soluble sugar content of IS decreased first and then increased, showing a significant increase of 37.5% at the pod-filling stage. Compared to the seedling stage, the soluble sugar content of IS significantly increased by 125% during the pod-filling stage. Compared to MP, the soluble sugar content significantly decreased in IP at the podding stage. Compared to MS, the soluble sugar content significantly increased by 30% in IS during the pod-filling stage. These results indicate that intercropping significantly increased the soluble sugar content in the root extracts of IS during the podding and pod-filling stages.

Comparing the four growth stages in the root exudates (Figure 2b), in IP, the soluble sugar content significantly increased by 186% during the pod-filling stage compared to the podding stage. Compared to the flowering-pegging stage, the soluble sugar content of IS significantly increased by 35% and 285% at the podding and pod-filling stages, respectively. Compared to MP and MS, both IP and IS showed a significant 32% increase in soluble sugar content during the pod-filling stage. These results indicate that intercropping increased the soluble sugar content in root exudates from IP and IS.

### 2.3. Effects of Sugarcane/Peanut Intercropping on the Amino Acid Content

Comparing the four growth stages in the root extracts (Figure 3a), the amino acid content of IP increased significantly at the flowering-pegging stage, four times higher than that at the seedling stage. The amino acid content of IS also significantly increased (five times) at the podding stage compared to the flowering-pegging stage. Compared to MP, the amino acid content at the seedling stage significantly decreased by 78% in IP. Compared to MS, the amino acid content of IS increased significantly, by 80%, during the podding stage, while it decreased significantly during the pod-filling stage. These results indicate that amino acid synthesis in the root extracts of IS was increased at the podding stage by intercropping, while at the pod-filling stages it was the opposite, and IP significantly decreased at the seeding.

In the four growth stages of the root exudates (Figure 3b), the amino acids of each treatment showed a trend of first increasing, then decreasing, and finally increasing again, with all treatments showing significant increases during the pod-filling stage. Compared to MP and IP, the amino acid content in both IP and IS did not show significant differences. These results indicate that intercropping had little effect on the amino acids in the root exudates of peanut and sugarcane.

### 2.4. Effects of Sugarcane/Peanut Intercropping on the Phenolic Acid Content

In the four growth stages of the root extracts (Figure 4a), the phenolic acid content of IP showed a trend of increasing, with a significant 87% increase during the pod-filling stage compared to the seedling stage. Similarly, in IS, the phenolic acid content significantly increased by 120% during the pod-filling stage compared to the podding stage. Compared to MP, the phenolic acid content was significantly reduced by 77% in IP at the seedling stage. Compared to MS, the phenolic acid content of IS was significantly reduced by 52% at the podding stage. These results indicate that sugarcane/peanut intercropping reduced the phenolic acid content in the root extracts of peanut at the seedling stage and in sugarcane at the podding stage.

In the root exudates (Figure 4b), the phenolic acid content of both IP and IS increased significantly during the pod-filling stage. Compared to MP and MS, IP and IS exhibited significantly increased phenolic acid content, with increases of 138% and 60%, respectively, during the pod-filling stage. These results indicate that sugarcane/peanut intercropping increased the phenolic acid content in the root exudates of peanut and sugarcane during the pod-filling stage.

### 2.5. Effects of Sugarcane/Peanut Intercropping on Rhizosphere Soil Enzyme Activity

The enzyme activity of rhizosphere soil was compared across four growth stages (Table 1). In IP, protease activity significantly decreased by 21% at the flowering-pegging stage compared to the seedling stage. In IS, protease activity initially decreased and then increased, with a significant 22% decrease at the podding stage compared to the seedling stage. The acid phosphatase activity of IP was significantly higher at the seedling stage than at the flowering-pegging, podding, and pod-filling stages, while in IS, it increased significantly during the pod-filling stage. The urease activity of IS during the pod-filling stage was 75% higher than that at the seedling stage and 46% higher than at the flowering-pegging stages. The catalase activity in both IP and IS showed a trend of increasing, decreasing, and then increasing again. The sucrase activity in IP and IS during the pod-filling stage was significantly lower than that at the podding stage.

In the rhizosphere soil, compared to MP (Table 1), the protease activity in IP decreased significantly during the pod-filling stage, and the acid phosphatase and urease activities in IP increased significantly at the podding and during the pod-filling stages by 16% and 57%, respectively. The catalase activity of IP significantly increased across all four stages. The sucrase activity of IP decreased significantly at the seedling and pod-filling stages. Compared to MS, the protease activity of IS significantly decreased at the podding and pod-filling stages. The urease activity of IS significantly decreased at the flowering-pegging stage and increased during the pod-filling stage, although the latter change was not significant. There were no significant differences in catalase activity. The sucrase activity significantly decreased at the seedling stage.

These results indicate that intercropping increased the acid phosphatase, urease, and catalase activities in the rhizosphere soil of IP during the podding or pod-filling stage, and also inhibited the protease and sucrose activities in the rhizosphere soil of both IP and IS at the seedling, podding, or pod-filling stages, but had little effect on the urease and catalase activities in sugarcane soil of IS.

### 2.6. Effects of Sugarcane/Peanut Intercropping on Rhizosphere Soil Nutrients

From the IP growth period, in the rhizosphere soil of IP (Table 2), the TN content increased significantly, by 1.5 times, at the flowering-pegging stage compared to the seedling stage. The TK and AP contents during the pod-filling stage were significantly higher than those at the podding stage. The change trend of TP in MP and IP showed a trend of first decreasing, then increasing and then decreasing. The TP content of MP and IP at the pod-filling stage was significantly decreased by 35% and 21% compared with that at the podding stage, respectively. The SOM content of MP and IP showed a decreasing trend. The SOM content of MP and IP at pod-filling stage was significantly decreased by 81% and 13% compared with that at the podding stage, respectively. It can be seen that intercropping can alleviate the decrease in TP and SOM content, and improve soil nutrients.

From the perspective of the IS growth period in rhizosphere soil of IS (Table 2), the TN, TP, and AN contents were higher at the podding and pod-filling stages than at the seedling and flowering-pegging stage stages. The TK content significantly increased at the flowering-pegging stage compared to the seedling stage. The AP content increased by 13% during the pod-filling stage compared to the podding stage. The SOM content of MS and IS first decreased and then increased, which increased by 29% and 67% at the pod-filling stage compared with the podding stage, respectively. The pH of both IP and IS at the flowering-pegging, podding, and during the pod-filling stages, was significantly higher than that at the seedling stage. The contents of TN, TP, AN, AP, and SOM in sugarcane increased at the podding or pod-filling stages.

Compared to MP (Table 2), the rhizosphere soil nutrients of IP were significantly different. TP, TK, AK and SOM of IP were significantly increased at the flowering-pegging, podding, and pod-filling stages, and TP content was significantly increased by 57%, 16% and 40%, respectively. TK content was significantly increased by 29%, 17%, and 14%, respectively. AK content was significantly increased by 106%, 107%, and 134%, respectively. SOM content was significantly increased by 111%, 55%, and 608%, respectively. AN content was significantly increased by 92% in podding. TN content was significantly increased by 165% in flowering-pegging, and the AP content had no significant difference. The pH value of IP significantly increased at the flowering-pegging and pod-filling stages. Sugarcane/peanut intercropping increased the TP, TK, AN, AK and SOM contents of IP in rhizosphere soil.

Compared to MS (Table 2), the TN content of IS significantly increased 65% at the flowering-pegging stage. The TP content of IS significantly increased 27% and 9% at the flowering-pegging, podding, respectively. while the TK content showed no significant difference. The AN content of IS increased significantly by 19% and 56% at the flowering-pegging and podding stages, respectively. The SOM content of IS increased significantly by 55% at the seedling stage, with no significant differences found at the podding and pod-filling stages. The pH of IS decreased significantly at the flowering-pegging stage. Intercropping increased TN and AN contents of IS in rhizosphere soil.

### 2.7. Effects of Sugarcane/Peanut Intercropping on Root Metabolites

A total of 349 metabolites were detected in MP, IP, MS, and IS, and they were divided into 58 categories: arboxylic acids and their derivatives (16%), fatty acyl groups (14%), benzene and its substituted derivatives (10%), and organic oxygen compounds (8%) (Figure 5a).

There were 29 distinct metabolites in MP and IP (Figure 5b), of which 11 were significantly upregulated and 18 were significantly downregulated. The differential metabolites of MP and IP were mainly carboxylic acids and their derivatives, fatty acids and conjugated, and benzene and substituted derivatives, including 6-phosphogluconic acid, 3-dehydroshikimic acid, fumaric acid, pentadecanoic acid, and lidocaine. Compared to MP, the 6-phosphogluconic acid, 3-dehydroshikimic acid, fumaric acid, pentadecanoic acid, and lidocaine contents increased 2.57–4.48 times in IP.

There were 23 different metabolites in MS and IS (Figure 5c), of which eight were significantly upregulated and 15 were significantly downregulated. The differential metabolites of MS and IS were mainly lipids, carboxylic acids and derivatives, and carbohydrates, including 10-hydroxycapric acid, clavulanate, aldose, and ribulose 1, 5-diphosphate. Compared to MS, the 10-hydroxycapric acid, clavulanate, aldose, and ribulose 1, 5-diphosphate contents increased 1.99–6.39 times in IS.

Pathway enrichment analysis of differential metabolites in MP and IP was performed using the Kyoto Encyclopedia of Genes and Genomes (KEGG) database. Differential metabolites were enriched in 23 pathways, and the first 20 pathways are shown in Figure 6a. Among them, the first five pathways with the highest degree of enrichment (the closer the *p*-value was to 0, the more significant the enrichment) were as follows: pyrimidine metabolism, alanine, aspartate, and glutamate metabolism, pantothenate and CoA biosynthesis, beta-alanine metabolism, and phenylalanine metabolism. Fumaric acid was involved in eight pathways, namely alanine, aspartate, and glutamate metabolism, phenylalanine metabolism, oxidative phosphorylation, the TCA cycle, arginine biosynthesis, pyruvate metabolism, fatty acid metabolism, and niacin and niacinamide metabolism (Appendix A).

Differential metabolites of MS and IS were enriched in 13 pathways (Figure 6b), and the first five pathways with the highest degree of enrichment were as follows: limonene and pinene degradation, carbon fixation in photosynthetic organisms, sphingolipid metabolism, valine, leucine, and isoleucine biosynthesis, and alanine, aspartate, and glutamate metabolism. According to the *p*-values and impact values from the KEGG results, the comparison between MP and IP showed significant differential metabolite enrichment in metabolic pathways, indicating that sugarcane/peanut intercropping has a great impact on peanut metabolism (Appendix A).

### 2.8. Correlation Analysis

The organic acid, soluble sugars, amino acids were significantly positively correlated with AP. Amino acids, soluble sugars were significantly positively correlated with AN. Catalase was significantly positively correlated with TK, AP and SOM (Table 3). Fumaric acid had a significantly positive correlation with catalase, AN, TP, and AK (Figure 7).

## 3. Materials and Methods

### 3.1. Test Site and Materials

This experiment was carried out in a field located in Lijian Town, Wuming District, Nanning City, Guangxi Zhuang Autonomous Region, at the Agricultural Science Institute (23°14′25″ N, 108°03′42″ E), and peanuts were planted at the front of the plot. The tested soils were acidic red earth, and the AN, AP, available potassium (AK), and SOM contents and pH values in the topsoil were 48.3 mg/kg, 3.7 mg/kg, 28.2 mg/kg, 15.7 g/kg, and 6.81, respectively.

The Guitang 42 variety of sugarcane and the Guihua 41 variety of peanut were provided by the Sugarcane Research Institute of Guangxi Zhuang Autonomous Region Academy of Agricultural Sciences, China. The sugarcane variety is one of the main cultivated varieties in Guangxi, and the peanut variety is a high yield and disease-resistant variety suitable for intercropping.

### 3.2. Experimental Design

Sugarcane and peanut were planted in a randomized design of a 6 m × 8 m pilot area on 17 March 2022, with three cropping treatments: monoculture sugarcane (MS), monoculture peanut (MP), and sugarcane/peanut intercropping (IS, IP), with three repeats for each treatment. MS and MP were used as controls. The layout was as follows: (1) Two rows of MP were planted on the ridge surface with a spacing of 0.6 m and width of 0.3 m, and each seed was dropped separately. Two rows of peanuts were planted on the ridge surface with a plant spacing of about 0.12 m, and each seed was placed separately. (2) MS was planted with a 1.2 m equal row spacing with 7 cane buds per meter. (3) Intercropping comprised two rows of sugarcane and four rows of peanuts. The row spacing between sugarcane and peanuts was 0.7 m, and the row spacing and plant spacing were the same as that of monoculture treatments. MS was used for the 1500 kg/ha compound fertilizer (N-P_2_O_5_-K_2_O = 15-15-15) for sugarcane and spread evenly in the planting ditch. For MP, 450 kg/ha compound fertilizer (N-P_2_O_5_-K_2_O = 15-15-15) and 750 kg/ha calcium magnesium phosphate (available P_2_O_5_ 18%) were applied evenly in the peanut planting ditch. IS and IP were fertilized separately according to the actual planting area. The amount of fertilizer was the same as that for MP.

### 3.3. Sample Collection and Determination

The rhizosphere soil samples were collected on 9 November 2022, 28 November 2022, 26 December 2022, and 16 January 2023, at the seedling, flowering-pegging, podding, and pod-filling stages, respectively, for peanuts. In this study, the four peanut growth stages during symbiosis were also used to describe sugarcane. 10 peanut and sugarcane plants were randomly dug from each treatment, gently shaken to remove excess bulk soil. The soil attached to the root system was regarded as rhizosphere soil and soil samples adhering to the roots were carefully collected with sterilized forceps and brushes, placed in sterile self-sealing bags, and transported to the laboratory. They were then air-dried, sieved, and used to determine the rhizosphere root soil nutrient content and enzyme activity. A total of 12 samples were collected for each stage, with three replicates per group, totaling 48 samples.

Root exudates and root extracts were collected and analyzed at the Experimental Building of the Guangxi Academy of Agricultural Sciences (22°50′50″ N, 108°14′45″ E). For root exudate collection, depending on the size of the peanut and sugarcane plants, peanuts (five plants) and sugarcane (three plants) were completely dug up for each treatment in each period. After, the impurities of the roots of peanuts and sugarcane were washed with distilled water and put into a plastic bucket containing 1000 mL of distilled water, respectively, with three repetitions for each treatment. After culturing for 2 h at 25 °C, the filtrate was filtered and collected, and then a small amount of thymol was added to inhibit microbial growth. The 1000 mL culture solution was divided into 200 mL and 800 mL, respectively. The 200 mL of filtered culture solution was concentrated to 25 mL at 40 °C by rotary evaporation (R1001-VN, Zhengzhou Greatwall Scientific Industrial and Trade Co., Ltd., Zhengzhou, China) [21], and the concentrated culture solution was used for the determination of organic acids, phenolic acids, amino acids, and soluble sugars, expressed in mg/L. The 800 mL culture solution was used to determine the effect of root exudates on soil nutrients. For root extract collection, 4 g of fresh roots was cleaned, cut into 1 cm pieces, and placed in a centrifuge tube. A volume of 40 mL of 95% ethanol was added, and samples were oscillated with ultrasonic waves for 30 min and centrifuged. The supernatant was collected and used for the determination of organic acids, phenolic acids, amino acids and soluble sugars, expressed in mg/L.

Total organic acids, amino acids, soluble sugar and phenolic acids content were determined by NaOH titration method [22], ninhydrin chromogenic method [23], anthrone colorimetric method [24], and Folin–Ciocalteu colorimetry [25], respectively.

Soil nutrients were determined according to Soil Agrochemical Analysis [26]. Total nitrogen (TN): weigh 1 g soil sample after air drying and grinding, add 8 mL concentrated H_2_SO_4_ and 2 g accelerator soil sample for digestion, wait for the digestion liquid and soil particles to turn gray and slightly green, and continue to digestion for 1 h, cool, and then be distilled. Two blank tests were performed while the sample was boiled. The distillate was collected and titrated with 0.01 mol/L H_2_SO_4_ standard solution. The distillate changed from blue-green to just reddish purple. Record the volume of the acid standard solution used.

Total phosphorus (TP): Weigh 0.25 g of ground soil sample and put it in the bottom of nickel (or silver) crucible. Add three drops of anhydrous ethanol and spread 2.0 g NaOH flat on the sample. Put the crucible into a high-temperature electric furnace and heat up. When the temperature rises to about 400 °C, cut off the power supply and suspend for 15 min. Then, continue to heat up to 720 °C, and keep for 15 min, take out slightly cold. Dissolve the frit with 10 mL of distilled water at about 80 °C, transfer it into a 100 mL volumetric bottle, and wash 10 mL 3 mol/L H_2_SO_4_ solution and water into the volumetric bottle several times, and cool, constant volume and filtration. Do a blank test at the same time. Absorb 5~10 mL of the sample solution to be tested into a 50 mL volumetric bottle and dilute it with water to about 30 mL. Add three drops of dinitrophenol indicator, and adjust the solution with 100 g/L Na_2_CO_3_ solution until just yellow. Add 5 mL Mo-Sb-Vc color development agent, shake well, and add distilled water to set volume. Leave for 30 min at room temperature above 20 °C. It was then measured with a spectrophotometer.

Total potassium (TK): Weigh 0.2 g of the air-dried soil sample, carefully place it in the bottom of the nickel (or silver) crucible, add five drops of anhydrous ethanol, and add 2 g NaOH on the surface of the sample. Put the crucible into a high-temperature electric furnace, and when the temperature rises to about 400 °C, cut off the power supply for 15 min to prevent the contents of the crucible from overflowing. Then, continue heating to 720 °C and maintain for 15 min. Remove and cool, add 10 mL of water at about 80 °C to dissolve the frit, transfer it to a 100 mL volumetric bottle, and wash the volumetric bottle with 10 mL of 3 mol/L H_2_SO_4_ solution and distilled water several times. After cooling, the volume is fixed, filtered, and then measured by a spectrophotometer.

Alkali-hydrolyzed nitrogen (AN): Weigh 2 g of air-dried and ground soil samples and 1 g FeSO_4_, and spread evenly and flatly in the outer chamber of the diffusion dish. Add 2 mL of 20 g/L H_3_BO_3_ solution in the inner chamber of the diffusion dish, cover the dish so that a slit is exposed in the outer chamber of the diffusion dish, then quickly add 10 mL of 1.8 mol/L NaOH solution in the outer chamber of the diffusion dish, and cover the dish. Horizontally gently rotate the diffusion dish, so that the NaOH solution and soil samples are fully mixed, placed in a constant temperature incubator at 40 °C heat preservation for 24 h. After the end of the incubation, remove the diffusion dish with 0.01 mol/L HCl standard solution of titrated ammonia absorbed in the inner chamber of the H_3_BO_3_; the color of the blue just changed to purplish red, that is, to reach the endpoint. A blank test was carried out at the same time as the sample determination to correct for reagent and titration errors. Calculations were made afterward.

Available phosphorus (AP): Weigh 2.50 g of ground and air-dried soil sample and place it in a 200 mL plastic bottle, add about 1 g of phosphorous free activated carbon, add 50 mL NaHCO_3_ extract, shake well and oscillate on an oscillator at 25 °C at a frequency of 180 r/min for 30 min, and filter it into a triangle bottle after shaking. Absorb 10 mL of filtrate into the colorimetric tube, slowly add 5 mL of color developer and slowly shake, add distilled water to the scale, and shake well. After being placed at room temperature for 30 min, colorimetric determination was performed at 700 nm with a spectrophotometer. Then, the calibration curve is drawn and calculated.

Available potassium (AK): Weigh 5 g of air-dried soil sample into 200 mL plastic bottle, add 50 mL CH_3_COONH_4_ solution, cover the bottle tightly and shake well, shake at 20–25 °C, 150–180 r/min for 30 min, and then filter. At the same time do blank test and draw calibration curve. Adjust the zero point of the instrument with CH_3_COONH_4_ solution, the filtrate is measured directly on the flame photometer, and then calculated.

SOM: Weigh 0.5 g of air-dried soil sample, put it into a hard test tube, then accurately add 10 mL 0.4 mol/L K_2_Cr_2_O_7_-H_2_SO_4_ solution from burette, shake well, and insert a glass funnel into the mouth of each test tube. Put the test tube in the oil bath at 185~190 °C for oil bath, after the oil bath temperature dropped to 170~180 °C, the solution boiled for 5 min and then put out the test tube to cool. Transfer the de-boiling liquid and the soil residue in the test tube into the 250 mL triangle bottle, rinse the test tube and small funnel with water, and merge the lotion into the triangle bottle, so that the total volume of the solution in the triangle bottle is controlled within 50~60 mL. The measured solution was titrated with FeSO_4_ standard solution by adding three drops of phenanthroline indicator. Run two blank tests simultaneously.

pH (H_2_O) were determined using an acidimeter (soil: water = 1:5), respectively. Soil enzyme activity was determined according to the methods described in Soil Enzymes and Methods for their Study [27].

For root metabolite analysis, root exudates were collected during the podding stage, concentrated to 25 mL (as described previously), and sent to Panomics Biomedical Technology Co., Ltd. in Suzhou City, Jiangsu Province, China, for analysis.

### 3.4. Data Analysis

Statistical analysis was conducted using IBM SPSS Statistics 26, and a statistically significant difference was considered at *p* < 0.05. Duncan’s test was used for post hoc examination. One-way analysis of variance (ANOVA) was used to examine the effects of intercropping on root extracts, root exudates, rhizosphere soil nutrients, and enzyme activities. The correlation between root extracts, exudates, metabolites, rhizosphere soil nutrients, and enzyme activities was assessed using Pearson’s correlation test. For metabolite data, the *p*-value was calculated according to a statistical test, and variable importance in projection (VIP) was calculated using the orthogonal partial least squares discriminant analysis (OPLS-DA) dimensionality reduction method. The differences between groups were calculated using the fold change to measure the influence strength and interpretation ability of each metabolite on sample classification and discrimination and to assist metabolite screening. When *p* < 0.05 and VIP > 1, the metabolite molecules were considered to be statistically significantly different.

## 4. Discussion

### 4.1. Response of Root Exudates to Sugarcane/Peanut Intercropping 

Compared to MP and MS, the organic acid content of IP increased, while that of IS decreased, indicating that intercropping changed the secretion characteristics of organic acids [28]. Compared to MS, the soluble sugar content in the root exudates of IS significantly decreased at the seedling, flowering-pegging, and podding stages but significantly increased during the pod-filling stage, and the amino acid content significantly increased at the podding stage. With plant growth, carboxylic acid is decomposed, decreasing the sugar content, while intercropping promotes the secretion of total sugars, organic acids, and amino acids in the roots [13,29]. The soluble sugars secreted by roots were significantly positively correlated with AN (Table 3), which may be due to the increase in N absorption by intercropping in both IP and IS, thereby increasing soluble sugar content during the pod-filling stage. Compared to MS, the phenolic acid content of both IP and IS decreased significantly at the seedling stage, while it significantly increased the phenolic acid content of IP during the pod-filling stage, which was inconsistent with a previous report from Qiu et al. [30]. Intercropping may alleviate allelopathic effects and lead to a decrease in the phenolic acid content [31], while the increase in the phenolic acid content of IP during the pod-filling stage may be due to peanut growth and development. Some phenolic acids are absorbed and accumulated by soil particles [32].

### 4.2. Response of Rhizosphere Soil Enzyme Activity to Sugarcane/Peanut Intercropping

The level of soil enzyme activity can reflect biological activity in the soil and the intensity of biochemical reactions, making it an important soil fertility index. Catalase is a significant oxidoreductase directly involved in the conversion of matter and energy in soil, and its activity may, to some extent, reflect the intensity of biological oxidation processes in soil. Compared to MS, the catalase activity in the rhizosphere soil of IP significantly increased, which was consistent with a previous report from Ma et al. [33]. This may be because the interaction between the roots of two crops not only accelerated organic matter conversion and enhanced biological oxidation metabolism but also changed the rhizosphere soil habitat and increased the number of microorganisms [34], changing the amount of enzymes that crop roots and microbes release into the soil [35]. Protease is the main enzyme involved in N mineral catalysis and the N cycle. Urease hydrolysis of urea can affect soil N metabolism, and sucrase is important for soil soluble nutrients. Compared to MP and MS, the protease and urease activities were significantly reduced in the rhizosphere soil of IS, and sucrase was reduced in the rhizosphere soil of IP. These results correspond to those of previous studies [36,37]. This may be due to the cross-superposition effect between roots under intercropping treatment, which results in abundant root secretions [38]. The mutually secreted substances inhibit the activities of related enzymes, but the specific reasons for this remain to be further studied.

### 4.3. Response of Rhizosphere Soil Nutrients to Sugarcane/Peanut Intercropping

The total content and availability of soil nutrients reflect the soil quality and directly influence plant growth and development. Compared to MP in the rhizosphere soil, the TP, TK, AN, AK, and SOM contents of IP significantly increased. Compared to MS, the TN and AN contents of IS significantly increased. Intercropping increased the rhizosphere soil nutrients, similar to the results of Zhang et al. [18] and Fu et al. [39], indicating that sugarcane/peanut intercropping improves the nutritional status of IP and IS in the rhizosphere. Studies have shown that when grasses and legumes are intercropped, grasses have more obvious intercropping advantages [40,41]. This study found that sugarcane/peanut intercropping had a greater impact on the soil nutrients in the rhizosphere soil of IP, and the TN, TP, AK, and SOM contents increased compared to MS. Compared to the podding stage, the contents of TP and SOM in both MP and IP were decreased at pod-filling stage, Because the critical period of dry matter accumulation and nutrient absorption is from the flowering-pegging stage to the podding stage of peanut [42,43], so the contents of TP and SOM were decreased at pod-filling stage. The TP contents of IP were higher than MP, and the decrease in TP contents in IP was smaller than that in MP at pod-filling stages; it may be that organic acids and soil enzyme activity lead to increased root phosphorus acquisition [18,44]. The SOM contents of IP changes were similar to those in TP, while the SOM of IS increased significantly during pod-filling stages, which was similar to the study of Lu et al. [45]. It may be that the biomass decomposition of litter increased under intercropping [46,47], and SOM was positively correlated with acid phosphatase and catalase, which led to the increase in TP and SOM contents and slowed down the decline. Leguminous crops may have a strong N fixation ability, thus increasing N sources in the soil, and non-leguminous crops may further stimulate and promote N fixation of peanut roots by competing for available N in the soil. Peanut is more sensitive to K and requires more K. K is directly connected with the chlorophyll content in plant leaves, and photosynthesis in chloroplasts is supported by higher K ion concentrations [48,49]. In this study, we found that the AK content of IP significantly increased at the flowering-pegging and podding stages. Peanut may require more photosynthetic energy to maintain growth during vigorous growth and development. Compared to MP, the soil pH of IP significantly increased at the flowering-pegging and pod-filling stages, inconsistent with the results of Zhang et al. [50]. This may be caused by the root decomposition of legumes, whose residues usually have excessive cation concentrations, coupled with decarboxylic organic anions and ammoniation of plant N residues, which results in an increase in pH value [51].

### 4.4. Response of Root Metabolites to Sugarcane/Peanut Intercropping

Organic acids exist widely in plants, and most are not only intermediate products of the tricarboxylic acid cycle but also intermediate products of the synthesis of sugars, amino acids, and lipids and play an important role in plant metabolism. Fumaric acid is a common organic acid in plants and is a component of the tricarboxylic acid cycle. Similar to starch and soluble sugars, these can be metabolized to produce energy and a carbon skeleton for the production of other compounds. Chia et al. [52] reported that the fumaric acid content in *Arabidopsis thaliana* exceeded the starch and soluble sugar contents. They speculated that fumaric acid might be a form of transient storage of binding carbon, such as starch and soluble sugar. As previously reported, the fumaric acid content increased significantly under wheat and broad bean intercropping [28]. This was consistent with the results of this study, indicating that intercropping affected organic acid secretion. The process of root secretion was thought to increase the activity of microorganisms and the exogenous enzymes they synthesize, thus accelerating the rate of C mineralization and N cycling in rhizosphere soil [53]. Meier [54] reported on simulated root exudate input and found that the input of organic acids (such as fumaric acid, malonic acid, oxalic acid, and citric acid) could increase N degradation through soil microorganisms and improve soil microbial biomass and microbial activity. Fumaric acid secreted by plants can activate inorganic phosphorus or organophosphorus in soil [55], and organic acids promote K decomposition and release through complexation and acidolysis [56]. Our data also showed that fumaric acid had a significantly positive correlation with catalase, AN, TP, and AK (Figure 7). Therefore, we speculated that fumaric acid increases enzyme activity, improves soil nutrients, and increases the N, P, and K contents in the rhizosphere soil. As previously reported, organic acids can promote the root system to secrete amino acids into the soil, and amino acids can partially replace nitrate N and manage N in plant root secretions [57,58]. Compared to MP, the fumaric acid content significantly increased in IP, and the amino acid content significantly increased in root exudates. Fumaric acid participates in the tricarboxylic acid cycle and in several metabolic pathways, such as alanine, aspartate, and glutamate metabolism. It significantly affected the alanine, aspartate, and glutamate metabolism and improved the soil N cycle in IP.

## 5. Conclusions

Sugarcane/peanut intercropping had a more significant effect on peanut than on sugarcane, especially during the podding and pod-filling stages. Intercropping can promote the release of soluble sugars, organic acids, amino acids, and phenolic acids from peanut roots and improve the activities of acid phosphatase, urease, and catalase in the rhizosphere soil. In addition, fumaric acid secreted by peanut roots during intercropping significantly affected alanine, aspartic acid, and glutamic acid metabolism, thereby improving the physical and chemical properties of soil and increasing the nutrients in rhizosphere soil. Therefore, reasonable intercropping can maximize its advantages, effectively improve the soil environment, and promote the healthy development of agricultural ecology.

## Figures and Tables

**Figure 1 plants-13-03257-f001:**
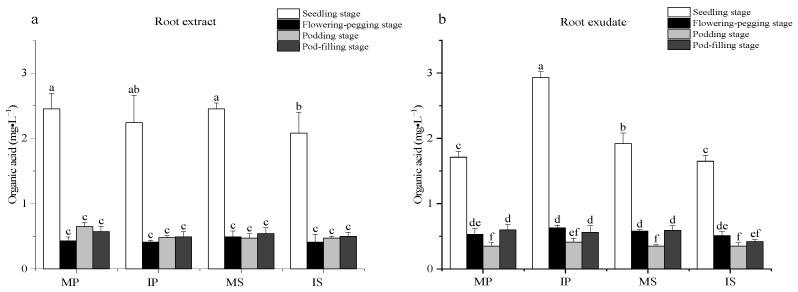
Effects of sugarcane/peanut intercropping on organic acids. (**a**) represents the change of organic acids in root extract under intercropping conditions, (**b**) represents the change of organic acids in root exudate under intercropping. Note: Different letters in the bar chart indicate significant differences (*p* < 0.05), the same below.

**Figure 2 plants-13-03257-f002:**
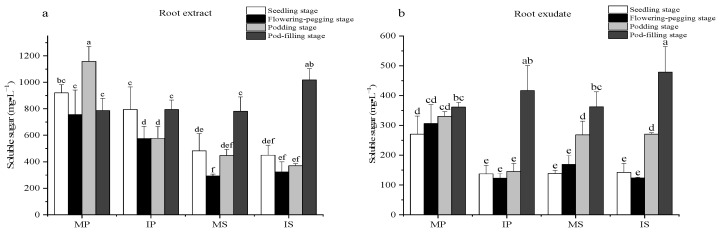
Effect of sugarcane/peanut intercropping on soluble sugar content. (**a**) shows the change of soluble sugar in root extract under intercropping conditions, (**b**) shows the change of soluble sugar in root exudate under intercropping.

**Figure 3 plants-13-03257-f003:**
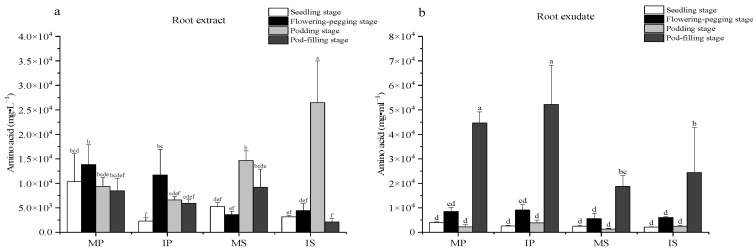
Effect of sugarcane/peanut intercropping on amino acids. (**a**) shows the change of amino acid in root extract under intercropping conditions, (**b**) shows the change of amino acid in root exudate under intercropping.

**Figure 4 plants-13-03257-f004:**
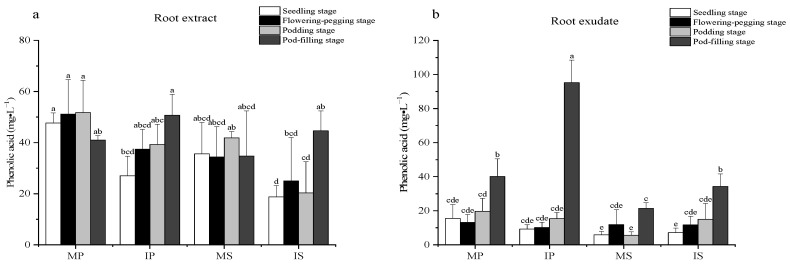
Effects of sugarcane/peanut intercropping on phenolic acids. (**a**) shows the change of phenolic acid in root extract under intercropping conditions, (**b**) shows the change of phenolic acid in root exudate under intercropping.

**Figure 5 plants-13-03257-f005:**
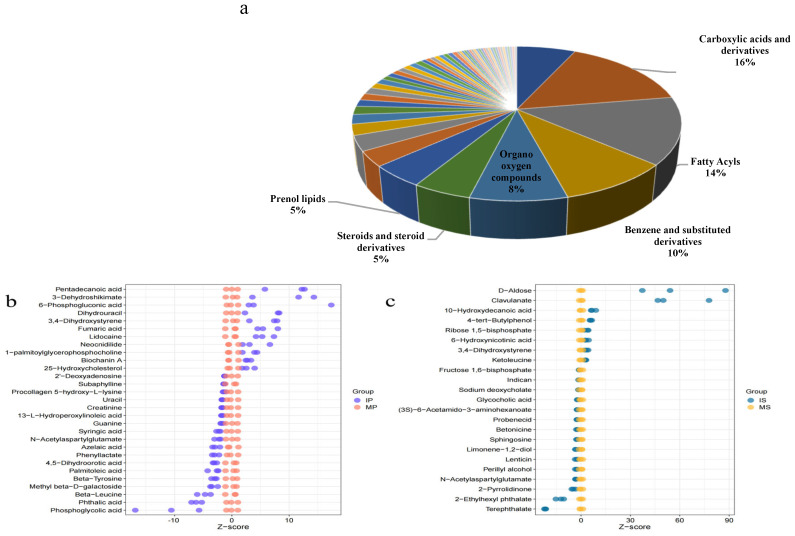
Effects of sugarcane/peanut intercropping on differential metabolites. Note: (**a**) shows the metabolites detected in monoculture peanut, intercrop peanut, monoculture sugarcane and intercrop sugarcane. (**b**) shows the trend and degree of difference of metabolites between MP and IP, (**c**) shows the trend and degree of difference of metabolites between MS and IS. In (**b**,**c**), the horizontal coordinate is the converted Z-score value of the relative content of metabolites in the sale, the vertical coordinate is the name of metabolites, and the color of the points represents different groups.

**Figure 6 plants-13-03257-f006:**
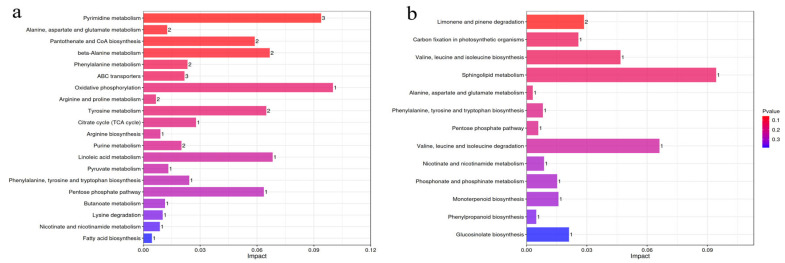
Effects of sugarcane/peanut intercropping on metabolic pathways. (**a**) shows enrichment of MP and IP metabolic pathways, (**b**) shows enrichment of MS and IS metabolic pathways. Note: the horizontal coordinate represents the Impact value that is enriched into different metabolic pathways, the vertical coordinate represents the metabolic pathway, and the number represents the corresponding number of metabolites on the pathway. Impact is the influence value of metabolic pathway, and the larger the impact of differential metabolites on the target pathway is. Color is correlated with the *p*-value, the redder the color, the smaller the *p*-value, the bluer the color, the larger the *p*-value, and the smaller the *p*-value, which means that the different metabolites have a more significant impact on this pathway.

**Figure 7 plants-13-03257-f007:**
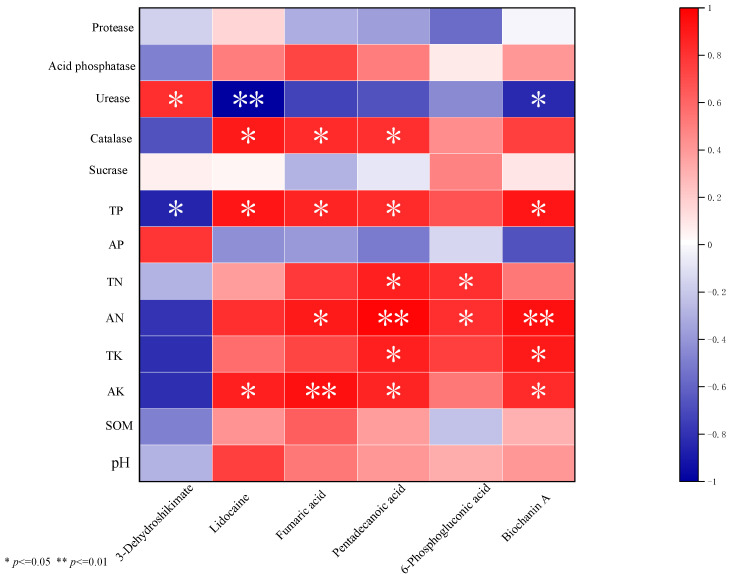
Correlation analysis of different metabolism and rhizosphere soil nutrients and enzyme activities of monoculture peanut and intercropping peanut.

**Table 1 plants-13-03257-t001:** Effects of enzimatic activity in rhizosphere in monoculture peanut (MP), monoculture sugarcane (MS), and sugarcane/peanut intercropping (IP, IS).

Treatment	Stage	Protease (mg·g^−1^)	Acid Phosphatase (mg·g^−1^)	Urease (mg·g^−1^)	Catalase (mg·g^−1^)	Catalase (mg·g^−1^)
MP	Seedling stage	0.78 ± 0.12 abcd	190.95 ± 6.53 b	3.79 ± 0.6 bcdef	75.53 ± 6.22 ef	9.94 ± 2.93 b
	Flowering-pegging stage	0.63 ± 0.06 efg	132.53 ± 11.31 efgh	3.85 ± 0.29 bcdef	87.39 ± 6.49 de	2.21 ± 1.33 f
	Podding stage	0.72 ± 0.05 bcdef	128.25 ± 18.63 ghi	4.41 ± 0.22 abcd	72.4 ± 11.9 f	6.81 ± 0.91 cd
	Pod-filling stage	0.74 ± 0.04 bcde	121.13 ± 6.53 hi	2.73 ± 0.6 f	77.54 ± 7.21 ef	5.87 ± 2.6 cd
IP	Seedling stage	0.87 ± 0.07 a	188.1 ± 8.55 bc	3.6 ± 0.43 cdef	95.51 ± 3.76 abcd	2.11 ± 0.9 f
	Flowering-pegging stage	0.68 ± 0.05 cdef	128.25 ± 11.31 ghi	4.66 ± 1.04 abcd	100.14 ± 6.72 abcd	1.61 ± 0.58 f
	Podding stage	0.71 ± 0.05 bcdef	149.63 ± 12.83 ef	3.29 ± 0.43 def	95.29 ± 4.37 abcd	6.77 ± 2.68 cd
	Pod-filling stage	0.61 ± 0.03 fg	128.25 ± 17.1 ghi	4.29 ± 0.56 abcd	102 ± 3.38 abc	0.89 ± 0.43 f
MS	Seedling stage	0.63 ± 0.02 efg	209.48 ± 8.55 a	4.04 ± 0.29 bcdef	98.42 ± 3.66 abcd	6.62 ± 0.75 cd
	Flowering-pegging stage	0.75 ± 0.1 bcd	173.14 ± 6.41 cd	5.48 ± 0.71 a	107.97 ± 0.34 a	5.64 ± 1.98 cde
	Podding stage	0.81 ± 0.07 ab	114 ± 4.94 i	4.1 ± 0.94 b cde	101.85 ± 2.11 abc	8.1 ± 1.35 bc
	Pod-filling stage	0.77 ± 0.06 abcd	138.23 ± 2.47 fgh	4.85 ± 0.99 abc	106.33 ± 1.37 ab	2.29 ± 0.59 f
IS	Seedling stage	0.72 ± 0.04 bcdef	218.03 ± 4.28 a	2.91 ± 0.39 ef	89.92 ± 5.26 cd	2.75 ± 2.56 ef
	Flowering-pegging stage	0.65 ± 0.03 defg	162.45 ± 8.55 de	3.48 ± 0.6 cdef	96.18 ± 5.5 abcd	3.92 ± 0.62 def
	Podding stage	0.56 ± 0.07 g	116.85 ± 6.53 i	3.79 ± 1.03 b cdef	90.97 ± 15.05 cd	16.85 ± 1.27 a
	Pod-filling stage	0.63 ± 0.06 efg	145.35 ± 8.55 efg	5.1 ± 1.15 ab	93.87 ± 10.03 bcd	3.95 ± 0.32 def

Note: average ± standard error, *n* = 3; Different lowercase letters after the same column of numbers indicate significant difference (*p* < 0.05).

**Table 2 plants-13-03257-t002:** Effects of sugarcane/peanut intercropping on rhizosphere soil nutrients.

Treatment	Stage	TN (g·kg^−1^)	TP (g·kg^−1^)	TK (g·kg^−1^)	AN (mg·kg^−1^)	AP (mg·kg^−1^)	AK (mg·kg^−1^)	SOM (g·kg^−1^)	pH
MP	Seedling stage	1.42 ± 0.11 cde	1.01 ± 0.05 h	199 ± 7.5 g	19.83 ± 4.55 g	24.58 ± 0.23 cde	61.3 ± 25.56 bc	135.02 ± 34.84 cdef	7.52 ± 0.09 g
	Flowering-pegging stage	1.34 ± 0.69 cde	0.57 ± 0.03 k	211.5 ± 8.66 fg	75.83 ± 13.25 cde	23.47 ± 0.32 efg	55.3 ± 21.34 c	74.34 ± 20.65 fg	7.82 ± 0.21 de
	Podding stage	0.24 ± 0.15 e	1.37 ± 0.02 c	199 ± 15 g	57.63 ± 1.76 def	23.85 ± 0.91 def	55.8 ± 12.49 c	66.75 ± 26.38 gh	7.78 ± 0.11 e
	Pod-filling stage	1.31 ± 1.35 cde	0.89 ± 0.05 i	226.5 ± 8.66 cdef	64.63 ± 16.95 def	24.72 ± 0.38 cde	43.8 ± 18.33 c	12.52 ± 1.14 h	7.85 ± 0.07 cde
IP	Seedling stage	1.42 ± 0.06 cde	1.05 ± 0.02 gh	214 ± 7.5 efg	11.9 ± 1.21 g	24.62 ± 0.18 cde	80.3 ± 16.9 abcd	172.95 ± 48.65 abcd	7.57 ± 0.07 fg
	Flowering-pegging stage	3.56 ± 0.73 a	0.9 ± 0.05 i	274 ± 7.5 a	65.33 ± 16.17 def	23.63 ± 0.14 ef	114.05 ± 6.75 a	157.02 ± 19.84 abcde	8.09 ± 0.06 ab
	Podding stage	0.6 ± 0.32 de	1.6 ± 0.04 b	234 ± 18.87 cde	110.83 ± 14.15 b	23.27 ± 0.42 fg	115.55 ± 6.75 a	103.16 ± 53.68 defg	7.92 ± 0.1 abcde
	Pod-filling stage	0.97 ± 0.66 cde	1.25 ± 0.04 d	259 ± 7.5 ab	52.03 ± 10.53 f	24.73 ± 0.4 cde	102.8 ± 35.27 a	88.75 ± 34.36 fg	8.11 ± 0.16 a
MS	Seedling stage	1.32 ± 0.07 cde	1.09 ± 0.01 fg	199 ± 0 g	10.03 ± 0.81 g	25.02 ± 0.18 cd	105.3 ± 16.04 a	137.3 ± 28.09 bcdef	7.19 ± 0.05 h
	Flowering-pegging stage	0.67 ± 0.28 de	0.86 ± 0.05 i	246.5 ± 11.46 bc	48.3 ± 4.59 f	22.38 ± 0.43 gh	54.8 ± 12 c	128.2 ± 34.08 cdefg	7.75 ± 0.1 ef
	Podding stage	1.86 ± 1.57 bcd	1.14 ± 0.03 ef	236.5 ± 19.84 cd	94.03 ± 13.9 bc	24.17 ± 0.13 cdef	101.8 ± 28.71 a	155.5 ± 36.29 abcde	7.91 ± 0.03 bcde
	Pod-filling stage	0.5 ± 0.03 de	2.07 ± 0.02 a	224 ± 8.66 d ef	79.33 ± 10.69 cd	26.2 ± 0.77 ab	113.3 ± 15.22 a	200.87 ± 35.94 ab	7.98 ± 0.04 abcd
IS	Seedling stage	1.43 ± 0.01 cde	0.72 ± 0.03 j	201.5 ± 4.33 g	10.97 ± 1.62 g	25.38 ± 0.26 bc	102.8 ± 30.11 a	213.16 ± 37.55 a	7.5 ± 0.01 g
	Flowering-pegging stage	2.19 ± 1.38 bc	1.1 ± 0.04 fg	261.5 ± 18.87 ab	57.63 ± 13.76 def	21.8 ± 1.26 h	80.3 ± 19.62 abcd	113.78 ± 31.86 cdefg	7.85 ± 0.05 cde
	Podding stage	3 ± 0.33 ab	1.25 ± 0.03 d	239 ± 4.33 cd	147 ± 24.25 a	23.57 ± 0.26 efg	98.8 ± 25.16 ab	97.1 ± 28.45 efg	7.74 ± 0.21 ef
	Pod-filling stage	0.56 ± 0.07 de	1.19 ± 0.02 e	219 ± 4.33 defg	103.13 ± 24.66 b	26.65 ± 1.78 a	107.3 ± 22.1 a	163.09 ± 41.16 abcd	8.04 ± 0.07 abc

Note: average ± standard error, *n* = 3; Different lowercase letters after the same column of numbers indicate significant difference (*p* < 0.05).

**Table 3 plants-13-03257-t003:** Correlation analysis of root extracts, root exudates, rhizosphere soil enzyme activities and soil nutrients under sugarcane/peanut intercropping conditions.

		TP	AP	TN	AN	TK	AP	SOM	pH
Root extract	organic acid	−0.224	0.292 *	−0.048	−0.721 **	−0.597 **	−0.027	0.341 *	−0.800 **
	soluble sugar	0.203	0.478 **	−0.406 **	−0.093	−0.378 **	−0.16	−0.109	0.175
	phenolic acid	0.024	0.1	−0.243	−0.011	−0.044	−0.227	−0.245	0.241
	Amino acids	0.052	−0.14	0.441 **	0.540 **	0.12	−0.007	−0.226	0.109
Root exudates	organic acids	−0.239	0.237	0.007	−0.734 **	−0.440 **	−0.021	0.349 *	−0.674 **
	soluble sugars	0.252	0.470 **	−0.315 *	0.362 *	−0.129	−0.12	−0.257	0.416 **
	phenolic acid	0.153	0.223	−0.251	0.075	0.268	0.03	−0.313 *	0.438 **
	amino acids	0.079	0.307 *	−0.192	0.091	0.244	−0.119	−0.362 *	0.431 **
protease		0.065	0.045	−0.192	−0.351 *	−0.211	−0.126	0.16	−0.079
Acid phosphatase		−0.249	0.123	−0.103	−0.740 **	−0.442 **	−0.001	0.456 **	−0.705 **
urease		0.2	0.036	−0.062	0.106	0.224	0.126	0.243	0.157
catalase		0.225	0	0.086	0.025	0.487 **	0.458 **	0.481 **	0.255
sucrase		0.098	−0.173	0.131	0.409 **	−0.145	−0.072	−0.24	−0.27

Note, * indicates *p* ≤ 0.05; ** indicates *p* ≤ 0.01.

## Data Availability

The data presented in this study are available on request from the corresponding author.

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
