# Peer review of "Effects of Sugarcane/Peanut Intercropping on Root Exudates and Rhizosphere Soil Nutrient"

_plants, 2024, doi:10.3390/plants13223257_

Round 1

Reviewer 1 Report

Comments and Suggestions for Authors

The study evaluates root exudates and soil nutrients from the rhizosphere in intercropping peanuts and sugar cane, compared to their respective monoculture cultivation. It is reported to be well-structured, with many data obtained that have been transformed into relevant information for understanding the chemical/biochemical transformations of the rhizosphere and soil. Improvements in the description of the experimental design are suggested, as well as greater bibliographic support in the discussion.

Author Response

Reviewer #1:

Comments 1: Improvements in the description of the experimental design are suggested, as well as greater bibliographic support in the discussion

Response: Thank you very much for your suggestions. I have revised the experimental design and added relevant references in the discussion.

Comments 2: Root exudates and Rhizosphere soil enzymes is only one reference in the text of the article. It is suggested that there are at least 3 references in the text.

Response:Thank you for your suggestions. Other references in this article also involve root exudates and rhizosphere soil enzyme activities, and this revision also adds and updates many related references. References added to the article have been highlighted in red.

Comments 3: available nitrogen (AN), ..., Suggestion: pH (H2O)

Response: Thanks for your suggestion, it has been modified, see section 2.3 Soil Nutrient Determination methods in the article.

Comments4: Suggestion:Specify the experimental design, e.g. randomized blocks? Indicate the number of replications/treatment, area of each replication/treatment

Response:Thanks for your suggestion, I have revised the chapter of experimental design, see Chapter 2.2 Experimental Design for details.

Comments 5: IS and IP? Are SI and PI just a field treatment, with collections near each crop?

Response: IS is intercropping sugarcane, IP is intercropping peanut, as described in 2.2 Experimental design, rhizosphere soil collection has now been described in detail.

Comments 6: suggestion:Indicate the fertilizer formulation, e.g. N:P:K...S

Response: Thank you for your suggestion, the compound fertilizer (N-P2O5-K2O=15-15-15) and calcium magnesium phosphate (available P2O5 18%) has been indicated in the article.

Comments 7: Suggetion: Just put a dot with normal formatting(μg.ml-1)

Response: Thanks for your suggestion, most references use "μg·ml-1", so this article uses "μg·ml-1".

Comments8: Suggetion:Put text before the table

Response: Thanks for your suggestion, the text has been adjusted to the front of the table.

Comments9: Suggestion:If you want to emphasise the contents in the various phenological stages, the data can be presented in lines(Fig 4)

Response: Thank you for your advice, I think your advice is very pertinent, but since we want the reader to see the changes in the different growing periods of sugarcane and peanuts, and the differences between monoculture and intercropping, we use the bar chart.

Comments 10: Suggestion:Put a space or a line before the reference to the next treatment.

Response: Thanks for your suggestion, the table has been modified, see Table 1 and Table 2 in the article for details.

Comments 11: Suggestion: Enzimatic activity in rhizosphere in monoculture peanut (MP), monoculture sugarcane (MS), and sugarcane/peanut intercropping (IP, IS).

Response: Thank you for your suggestions, which have been revised under Table 1 headings in Section 3.5.

Comments 12: Put a space or a line before the reference to the next treatment

Response:Thank you for your suggestion. In order to help readers read the form better, the form format has been modified.

Comments13:Suggestion: Put a space between the value and letters (P<0.05),lower case.

Response:Thanks for your suggestion, it has been modified.

Comments14: this “ K ” macronutrient.

Response: K here includes K of other forms.

Reviewer 2 Report

Comments and Suggestions for Authors

This is a paper which may be published but it shows severe deficiency. Also the authors did not work carefully on this ms. Citations and references in the ms are not prepared according to the Plants rules.

Intro

Quantitative role of sugarcane/peanut intercropping in the considered region?

Math. Meth.

Describe the determination of AN, AP, AK and SOM in detail, also the dtermination of the total concentrations which are mentioned in results but not here.

"MS was mixed with 1500 kg/ha fertilizer" What does this sentence mean?? What is compound fertilizer?Which rates of N, P, K/ha were applied?

Root exudation into distilled water may disturb the roots because Ca is eluted. Describe in detaiol the rhizosphere soil collection, the collection and determination of root exudates. The descriptions here are highly unscientific!

Results

Fig. 1. What does the unit mg/ml mean,which acids were determined and summarized?

Critics on fig. 2,3, are similar.

Table 2. Do you relly think that TP (total P? nowhere explained!)and SOM will change by a factor of two in the rhizosphere soil during the experiment? pH not PH.

The discussion and the conclusions drawn from theses analyses and results are not relevant because of metholodical failures and results which are not probable.

Comments on the Quality of English Language

see above

Author Response

Reviewer #2:

Comments 1:This is a paper which may be published but it shows severe deficiency. Also the authors did not work carefully on this ms. Citations and references in the ms are not prepared according to the Plants rules.

Response:Thanks for your correction, the introduction and references of the article have been revised and replaced, and the modified part has been marked in red.

Comments 2: Quantitative role of sugarcane/peanut intercropping in the considered region?

Response: Thank you for your correction. I'm sorry for the inadequacy of the previous introduction, which put too much emphasis on quantity. Now we have revised the introduction, see article for details.

Comments 3: Describe the determination of AN, AP, AK and SOM in detail, also the dtermination of the total concentrations which are mentioned in results but not here.

Response: Thanks for your correction, the measurement method of soil nutrients and total concentration has been corrected, see 2.3. Sample collection and determination in the article.

Comments 4: "MS was mixed with 1500 kg/ha fertilizer" What does this sentence mean?? What is compound fertilizer?Which rates of N, P, K/ha were applied?

Response: Thanks for your correction, the expression of fertilizer has been modified to “ MS was used 1500 kg/ha compound fertilizer (N-P2O5-K2O=15-15-15) for sugarcane and spread evenly in planting ditch. For MP, 450 kg/ha compound fertilizer (N-P2O5-K2O=15-15-15) and 750 kg/ha calcium magnesium phosphate (available P2O5 18%)were applied evenly in the peanut planting ditch.”

Comments 5: Root exudation into distilled water may disturb the roots because Ca is eluted. Describe in detaiol the rhizosphere soil collection, the collection and determination of root exudates. The descriptions here are highly unscientific!

Response: Thank you for your reminder, Root exudates mainly include low molecular weight (organic acids, sugars, phenols and various amino acids, etc.) and high molecular weight organic compounds (proteins, mucus, etc.), so the extraction of root exudates in this study was not interfered with by Ca. The description of root soil collection and root exudate extraction has been revised.

Comments 6: Fig. 1. What does the unit mg/ml mean,which acids were determined and summarized?

Response: “mg/ml” means how many mg of total organic acids are contained in 1 mL concentrated root exudate. In this paper, organic acids are determined to contain total organic acids

Critics on fig. 2,3, are similar.

Comments 7: Table 2. Do you relly think that TP (total P? nowhere explained!)and SOM will change by a factor of two in the rhizosphere soil during the experiment? pH not PH.

Response: I am very sorry that the result of Table 2 is not described clearly. In the original material and method, there is an explanation of "total phosphorus (TP)" . Now we have modified the result description of Table 2 and summarized the results of different periods and different treatments, for example, “In general, sugarcane/peanut intercropping increased the TP, TK, AN, AK and SOM contents of IP in rhizosphere soil”, “In general, intercropping increased TN and AN contents of IS in rhizosphere soil”, “In general, the rhizosphere soil nutrients of IP increased at the podding or pod-filling stages”, “In general, the contents of TN, TP, AN and AP in sugarcane increased at at the podding or pod-filling stages”.

Comments 8: The discussion and the conclusions drawn from theses analyses and results are not relevant because of metholodical failures and results which are not probable.

Response: I am very sorry that I did not describe the “materials and methods” of this study clearly, which caused such misunderstanding. The extraction of root exudates and the determination of indicators in this experiment were carried out with reference to previous studies, and now we have improved the material and method sections, as well as modifying some of the descriptions of the results such as the description of the results in Table 2, which are detailed in the article.

Reviewer 3 Report

Comments and Suggestions for Authors

The manuscript evaluated the effects of sugarcane/peanut intercropping on root exudates and rhizosphere soil nutrient. A lot of data was presented, and the results offer a theoretical foundation for further exploration into achieving high quality and high yields of sugarcane and peanut through intercropping. However, several issues need to be addressed before publication.

1. Abstract: Please explain “The effects of intercropping differed in peanut and sugarcane at different growth stages”. For example, authors could mention the critical growth stage.

2. Introduction: Current knowedgement gap was not pointed out. The novelty should be strengthened.

3. Materials and methods: The setup of the intercropping treatment should be refered.

4. Results and analysis: Sugarcane and peanut yield should be provided and discussed.

5. Discussion: The sequence number of subtitles should be rearranged.

Comments on the Quality of English Language

The language should be polised by native English speakers.

Author Response

Reviewer#3:

The manuscript evaluated the effects of sugarcane/peanut intercropping on root exudates and rhizosphere soil nutrient. A lot of data was presented, and the results offer a theoretical foundation for further exploration into achieving high quality and high yields of sugarcane and peanut through intercropping. However, several issues need to be addressed before publication.

Comments 1:. Abstract: Please explain “The effects of intercropping differed in peanut and sugarcane at different growth stages”. For example, authors could mention the critical growth stage.

Response: Thank you for your suggestion, it has been revised in the abstract to "The effects of intercropping differed in peanut and sugarcane at different growth stages, and the difference between podding and pod-filling stages was significant”.

Comments 2: Introduction: Current knowedgement gap was not pointed out. The novelty should be strengthened.

Response: Thank you for your suggestion that gaps in current relevant research have been mentioned in the abstract “ In summary, the current research on intercropping focuses more on soil nutrient changes and crop yield improvement, and the mechanism of correlation between root secretion and soil nutrients is still unclear ”.

Comments 3: Materials and methods: The setup of the intercropping treatment should be refered.

Response: Thank you for your suggestion, which is explained in detail in 2.2  Experimental design, “Intercropping comprised two rows of sugarcane and four rows of peanuts. The row spacing between sugarcane and peanuts was 0.7 m, and the row spacing and plant spacing were the same as that of monoculture treatments.”

Comments 4:. Results and analysis: Sugarcane and peanut yield should be provided and discussed.

Response: Thank you for your advice, which I find very pertinent, but this article focuses on intercropping on root exudates, soil and their interactions. The effect of intercropping on yield will be discussed in the next article.

Comments 5:. Discussion: The sequence number of subtitles should be rearranged.

Response: Thanks for your correction, it has been modified.

Round 2

Reviewer 2 Report

Comments and Suggestions for Authors

Again total P should change in the rhizosphere (as a result of root activity?).

The authors are obvipusly not aware about the term total P. Also the changes in O.M in the rhizosphere are rather "astonishing". A concentration of mg/ml of exudates fully depends on the ratio of root surface versus solution and give no information on the root release of organic compounds. 

The authors even did not understand my critics on their first version. I am also astonished that despite my recommendation "reject" I recieved a slightly corrected paper. 

Author Response

Dear Reviewer,

Thank you very much for your comments. We deeply regret that we did not fully understand your suggestions in the first version. Those comments are all valuable and very helpful for revising and improving our paper (ID:3135238), as well as the important guiding significance to our researches.We have read comments carefully and have made correction which we hope meet with approval. The modified parts of the paper are marked in blue.

Again total P should change in the rhizosphere (as a result of root activity?).The authors are obviously not aware about the term total P.  Also the changes in O.M in the rhizosphere are rather "astonishing".

Response: The changes of TP and SOM content at different stages and the reasons has been added in the result analyze and discussion. The change of TP content in MP and IP showed a trend of first decreasing, then increasing and then decreasing. The TP content of MP and IP at pod-filling stage was significantly decreased by 35% and 21% compared with that at podding stage, respectively. 

The SOM content of MP and IP showed a decreasing trend with the extension of the growth period. The SOM content of MP and IP at pod-filling stage was significantly decreased by 81% and 13% compared with that at podding stage, respectively. It can be seen that intercropping can alleviate the decrease of TP and SOM content, and improve soil nutrients. 

The SOM content of MS and IS first decreased and then increased, which increased by 29% and 67% at pod-filling stage compared with podding stage, respectively.

A concentration of mg/ml of exudates fully depends on the ratio of root surface versus solution and give no information on the root release of organic compounds. 

Response: The concentration of root exudates and the measured organic compounds are described in “2.2. Experimental design”. “Depending on the size of the peanut and sugarcane plants, peanuts (five plants) and sugarcane (three plants) were completely dug up for each treatment in each period. After the impurities of the roots of peanuts and sugarcane were washed with distilled water and put into a plastic bucket containing 1000 mL of distilled water for 2 h at 25°C. Then the 1000 mL culture solution was divided into 200 mL and 800 mL respectively. The 200 mL of filtered culture solution was concentrated to 25 mL at 40°C by R1001-VN rotary evaporation (Tian et al., 2003), and the concentrated culture solution was used for the determination of organic acids, phenolic acids, amino acids and soluble sugars, expressed in mg/L.” The 800 mL culture solution were used to research the effect of root exudates on soil nutrients.